# A WIDELY USED PROTOCOL FOR EEG CLASSIFICATION EXPERIMENTS LEADS TO A CONFOUND

## ABSTRACT

A temporal confound has been previously reported in a (now widely used) dataset that others have tried to suggest is nonetheless justified. Despite attempts to make the community aware of this confound, a significant number of publications continue to use the confounded dataset, thereby drawing unsupported conclusions. We present a new experiment that conclusively demonstrates that the identified confound in the dataset cannot be explained away by recourse to factors such as block design, session duration, number of subjects, or pooling multiple subjects. We advise caution when designing, conducting, and interpreting the results of experiments that use this problematic protocol.

## 1 INTRODUCTION

Recent work (Spampinato et al., 2017; Palazzo et al., 2017; Kavasidis et al., 2017; Palazzo et al., 2018; 2020a;b; 2021; 2024) claimed to decode the image class of stimuli presented to subjects undergoing electroencephalography (EEG) from the data recorded for individual stimuli, with extremely high accuracy. Follow-on work (Li et al., 2021; Ahmed et al., 2021; 2022; Bharadwaj et al., 2023) called these claims into question. Because all EEG exhibits drift, a clock is essentially encoded into the signal. The dataset used by Spampinato et al. (2017), Kavasidis et al. (2017), and Palazzo et al. (2017; 2018; 2020a;b; 2021; 2024) employed a protocol whereby all stimuli of a given class were presented temporally adjacent in a single block. The training and test samples for a given class were taken from this same block. This introduces a temporal confound: stimulus class is correlated with time. It is thus unclear whether the classifier is decoding stimulus class or the embedded clock due to drift. When Li et al. (2021), Ahmed et al. (2021; 2022), and Bharadwaj et al. (2023) replicated the experiment with randomized stimulus order, classification accuracy dropped to essentially chance.

Recent responses (Palazzo et al., 2020b; 2024) raise several issues which they claim invalidate the claims of Li et al. (2021), Ahmed et al. (2021; 2022), and Bharadwaj et al. (2023).

I) Block designs are to be preferred over interleaved (*i.e.*, randomized) designs in order to achieve higher signal-to-noise ratio, to allow the signal from one trial to bleed into an adjacent trial without impacting stimulus class, to motivate subjects to focus on the similarities between adjacent trials rather than their differences, to make the class salient, and to compensate for potential subject inattentiveness.

II) Experimental sessions should be kept short to avoid subject fatigue and to mitigate any potential temporal confound.

III) Results in Li et al. (2021), Ahmed et al. (2021; 2022), and Bharadwaj et al. (2023) are invalid because key results are only presented on data from one subject.

IV) Analyses that pool data from multiple subjects are to be preferred over analyses of data from a single-subject, to mitigate greater within-subject and lesser cross-subject similarity of the EEG signal and to reduce the bias due to possible temporal correlations that may exist in a single subject's neural responses.

Here, we counter these claims with a new data collection that exactly mirrors that of Spampinato et al. (2017) to repeat the key experiments from Li et al. (2021), Ahmed et al. (2021; 2022), and Bharadwaj et al. (2023) in order to address all of the above issues; nonetheless a temporal confound in their data still exists. Following Spampinato et al. (2017), we employ a block design where all stimuli of a given class are presented as temporally adjacent trials in a single block and adopt their exact timing so as to keep all sessions the same length as theirs. In addition, we collect data from

six different subjects, and are therefore able to present results both on individual subjects and when pooling all six subjects. Central to this new experiment is the extension of an idea, first presented in Li et al. (2021), whereby all subjects participate in session groups with two different block orders. Whereas Li et al. (2021) did this only for one of the six subjects (subject 6), here we do this for all six subjects.

The key finding is that classification accuracy is only high when training and test samples are taken from the same blocks of the same session group, as was done by Spampinato et al. (2017), Kavasidis et al. (2017), and Palazzo et al. (2017; 2018; 2020a;b; 2021; 2024). However when training samples are taken from the blocks of one session group and the test samples are taken from the blocks of a different session group with a different block order, classification accuracy drops precipitously. Since the only difference between these two conditions is temporal ordering of the classes, this conclusively demonstrates that the results of Spampinato et al. (2017), Kavasidis et al. (2017), and Palazzo et al. (2017; 2018; 2020a;b; 2021; 2024) are not due, as they claim, to the block design, short experimental sessions, data from six subjects, or pooling multiple subjects, but rather support our point that their results are due to the temporal confound.

## 2 SIGNIFICANCE

Several independent lines of research have refuted a large body of flawed work (Spampinato et al., 2017; Kavasidis et al., 2017; Palazzo et al., 2017; 2018; 2020a;b; 2021; 2024) along completely different axes. Li et al. (2021) demonstrated that the dataset used (Spampinato et al., 2017), and the methods used to collect that dataset, suffer from a temporal confound, correlating stimulus class with experiment timing. Accuracy drops to chance when the confound is removed. Ahmed et al. (2021) demonstrated that this holds even with a much larger dataset. Ahmed et al. (2022) demonstrated that this holds for the additional classifiers used in Palazzo et al. (2018; 2020a;b; 2021). Bharadwaj et al. (2023) demonstrated that this holds even when using supertrials.

Here we progress beyond prior demonstrations that the particular dataset (Spampinato et al., 2017) is confounded. We respond specifically to claims I—IV above in ways that have not previously been addressed. This is significant because the confounded dataset continues to be used, and confounded collection methods continue to be used to collect new confounded datasets, justified, in part, by claims I—IV. This is further significant for the following reasons:

- Nearly one hundred papers (An & Cho, 2016; Spampinato et al., 2016; Ben Said et al., 2017; Bozal Chaves, 2017; Kavasidis et al., 2017; Palazzo et al., 2017; Parekh et al., 2017; Spampinato et al., 2017; Zhang et al., 2017; Du et al., 2018; Fares et al., 2018; Kumar et al., 2018; Palazzo et al., 2018; Piplani et al., 2018; Tirupattur et al., 2018; Wang et al., 2018; Zhang & Liu, 2018; Zhang et al., 2018; Zhong et al., 2018; Du et al., 2019; Hwang et al., 2019; Jiang et al., 2019; Jiao et al., 2019; Long et al., 2019; Mukherjee et al., 2019; Uys, 2019; Wang et al., 2019; Cudlenco et al., 2020; Fares et al., 2020; Li et al., 2020; Palazzo et al., 2020a;b; Wang et al., 2020; Zheng et al., 2020a;b; Palazzo et al., 2021; Zheng & Chen, 2021; Ma et al., 2021; Mo et al., 2021; Jiang et al., 2021; Lee et al., 2021; Cavazza et al., 2022; Khaleghi et al., 2022; Lee et al., 2022; Mishra et al., 2022; Mishra, 2022; Scharnagl & Groth, 2022; Shimizu & Srinivasan, 2022; Ahmadieh et al., 2023; Bai et al., 2023; Du et al., 2023; Duan et al., 2023; Hasan & A, 2023; Imani et al., 2023; Lan et al., 2023; Lee et al., 2023; Liu et al., 2023; Singh et al., 2023; Song et al., 2023; Wahengbam et al., 2023; Zeng et al., 2023b;a; Fan et al., 2024; Ferrante et al., 2024a;b; Gou et al., 2024; Lei et al., 2024; Liu et al., 2024a;b; Luvsansambuu et al., 2024; Mishra et al., 2024; Mwata-Velu et al., 2024; Ngo et al., 2024; Palazzo et al., 2024; Pan et al., 2024; Qian et al., 2024; Singh et al., 2024; Tang et al., 2024; de la Torre-Ortiz et al., 2024; Yang & Liu, 2024; Ye et al., 2024; Zheng et al., 2024b;a; Zhu et al., 2024; Deng et al., 2025; Fares, 2025; Fu et al., 2025; Lopez et al., 2025; Mehmood et al., 2025; Singh et al., 2025; Xiang et al., 2025) draw flawed conclusions based on the confounded dataset from Spampinato et al. (2017) and datasets suffering from the same confound.
- A number of new datasets have been collected with this same confounded protocol (Gou et al., 2024; Pan et al., 2024; Zhu et al., 2024; Qian et al., 2024; Uys, 2019; Shimizu & Srinivasan, 2022; Liu et al., 2024b; Wang et al., 2019; 2020; Ma et al., 2021; Cudlenco

et al., 2020; Zheng et al., 2024b; Cavazza et al., 2022; Luvsansambuu et al., 2024; Liu et al., 2023; Bai et al., 2023; Parekh et al., 2017).

- A number of these have been publicly released and are used by others. For example, Singh et al. (2023), Singh et al. (2024), and Lopez et al. (2025) use the dataset reported in Kumar et al. (2018) and Duan et al. (2023), Singh et al. (2024), and Lopez et al. (2025) use the dataset reported in Ma et al. (2021).
- This is further egregious because Palazzo et al. (2020b; 2024) continue to claim that their dataset (Spampinato et al., 2017), and their results that were obtained with that dataset (Spampinato et al., 2017; Kavasidis et al., 2017; Palazzo et al., 2017; 2018; 2020a;b; 2021; 2024), are valid, despite the refutations in Li et al. (2021), Ahmed et al. (2021; 2022), and Bharadwaj et al. (2023), in part, because of claims I—IV in Palazzo et al. (2020b; 2024).
- This has been used to justify continued publication of a huge body of flawed work based on confounded datasets (Cavazza et al., 2022; Khaleghi et al., 2022; Lee et al., 2022; Mishra et al., 2022; Mishra, 2022; Scharnagl & Groth, 2022; Shimizu & Srinivasan, 2022; Ahmadieh et al., 2023; Bai et al., 2023; Du et al., 2023; Duan et al., 2023; Hasan & A, 2023; Imani et al., 2023; Lan et al., 2023; Lee et al., 2023; Liu et al., 2023; Singh et al., 2023; Song et al., 2023; Wahengbam et al., 2023; Zeng et al., 2023b;a; Fan et al., 2024; Ferrante et al., 2024a;b; Gou et al., 2024; Lei et al., 2024; Liu et al., 2024a;b; Luvsansambuu et al., 2024; Mishra et al., 2024; Mwata-Velu et al., 2024; Ngo et al., 2024; Palazzo et al., 2024; Pan et al., 2024; Qian et al., 2024; Singh et al., 2024; Tang et al., 2024; de la Torre-Ortiz et al., 2024; Yang & Liu, 2024; Ye et al., 2024; Zheng et al., 2024b;a; Zhu et al., 2024; Deng et al., 2025; Fares, 2025; Fu et al., 2025; Lopez et al., 2025; Mehmood et al., 2025; Singh et al., 2025; Xiang et al., 2025) even after the confound became known through the work of Li et al. (2021), Ahmed et al. (2021; 2022), and Bharadwaj et al. (2023).

Current machine-learning conferences, and more generally, computer-science conferences and journals, are loathe to publish refutations. Observing this, Schaeffer et al. (2025) proposed that the field of machine-learning establish a "refutations and critiques" track in prominent conferences. While we applaud and support this proposal, the current lack of such a track should not be an impediment to publishing refutations. Scientific journals in other fields have long done so, often resulting in retraction of flawed work. Schaeffer et al. (2025) offer five example pieces of claimed flawed work in machine learning. Each is an individual paper. These pale in comparison to the flaws we uncover here: a systemic flaw of the entire peer review process across an entire field of inquiry, namely classification of stimulus image class from EEG recordings, that affects seventeen datasets and ninety one papers. Moreover, none of the five examples in Schaeffer et al. (2025) are egregious; here the authors of the flawed work continue to argue for its validity despite four refereed refutations and fifty new flawed papers have been published subsequent to these four refereed refutations. This argues for the need to make the community aware of the severity of the issue.

## 3 METHOD

We followed the protocol from Spampinato et al. (2017, Table 1). Six subjects each underwent eight sessions. Each session presented the stimuli for 10 out of the 40 classes. Each session presented all 50 stimuli for a given class as a block. Each stimulus was presented for 0.5 s, with no blanking between stimuli, and 10 s of blanking at the end of each block. Spampinato et al. (2017)'s original protocol (apparently 5 minutes and 50 s) appears not to have included any blanking at the beginning of each session; however, we included 10 s of blanking at the beginning of each session. Thus presenting each class took 35 s and presenting all 10 classes in each session took 360 s, *i.e.*, 6 minutes (not as Palazzo et al. 2020b; 2024 claim "about 4 minutes"). Following Palazzo et al. (2020b), "After each session the subjects had time to rest and they continued the experiment whenever they felt ready." Not including rest time between sessions, the whole experiment took 48 minutes per subject, broken up into eight 6 minute sessions.

The eight sessions per subject were divided into two groups of four sessions each. Each group of four sessions presented all 50 stimuli of all 40 classes, *i.e.*, 2000 stimuli, split into 50 stimuli per block, all of the same class, with 10 blocks per session and arbitrary subject-decided rest between sessions. Group A, the first four sessions for each subject, presented the classes and the stimuli within each class for all subjects in the same order. Thus all six subjects were presented all 2000 stimuli in group A in the same order. Group B, the second four sessions for each subject, randomized the class

Table 1: Average classification accuracies on the test sets, averaged across all folds. Stars denote statistically significant above-chance classification accuracy by a binomial cmf ($p < 0.005$).

| | Condition | | | | | | | |
| | LSTM | | | | EEGChannelNet | | | |
| Subject | I | II | III | IV | I | II | III | IV |
|---|---|---|---|---|---|---|---|---|
| 1 | 88.2%* | 91.0%* | 3.1% | 1.6% | 94.7%* | 96.4%* | 1.6% | 1.8% |
| 2 | 85.5%* | 92.9%* | 3.2% | 2.2% | 95.1%* | 95.3%* | 2.5% | 2.2% |
| 3 | 89.7%* | 89.8%* | 1.0% | 1.9% | 93.9%* | 88.4%* | 2.2% | 2.5% |
| 4 | 85.2%* | 87.5%* | 2.8% | 3.8%* | 81.9%* | 94.3%* | 2.9% | 1.4% |
| 5 | 83.4%* | 92.5%* | 2.9% | 2.6% | 97.1%* | 96.8%* | 4.1%* | 2.5% |
| 6 | 67.9%* | 72.5%* | 1.4% | 1.7% | 93.1%* | 94.9%* | 1.5% | 1.5% |
| pooled | 84.5%* | 85.6%* | 2.3% | 2.6% | 95.3%* | 95.1%* | 2.9%* | 2.5% |

and stimulus order by subject but kept all stimuli for a given class temporally adjacent in the same block. Thus all six subjects were presented all 2000 stimuli in group B in order that differed from each other and from group A. The remainder of the protocol was as described by Li et al. (2021) and Ahmed et al. (2022), which is similar to the protocol described by Ahmed et al. (2021) and Bharadwaj et al. (2023).

EEG data was recorded with a Magstim EGI (128 channels). Eye tracking data was recorded with an Eyelink 1000 Plus and used to confirm subject attention.

# 4 ANALYSIS

Following Spampinato et al. (2017), all data was preprocessed with a 59-61 Hz notch filter and a 14-71 Hz bandpass filter. Further all analyses were performed on 440 ms windows starting 40 ms after stimulus onset. We performed five-fold cross validation, partitioning the data for each group for each subject into five disjoint covering portions, each containing exactly 20% of the trials for each class. In each fold of the analysis, four of the five portions (80%) were taken as a training set and the remaining portion (20%) was taken as a test set, rotating among the portion taken as test set. To avoid the repeated-stimulus confound (Kilgallen et al., 2024a;b; 2025), portions were constructed so that corresponding portions for each group and each subject contained recordings for the same stimuli. This ensured that when we performed pooled-subject analyses and cross-group analyses within subject, no recordings from the same stimulus image appeared in both the training and test sets.

We performed two kinds of analyses. First, we performed independent within-subject analyses on each of the six subjects. Second, we performed an analysis that pooled the data from all six subjects.

Analyses for four conditions were performed:

**Condition I:** Train on the training set for group A, test on the test set for group A.
**Condition II:** Train on the training set for group B, test on the test set for group B.
**Condition III:** Train on the training set for group A, test on the test set for group B.
**Condition IV:** Train on the training set for group B, test on the test set for group A.

Each analysis was done both on the LSTM from Spampinato et al. (2017) and on EEGChannelNet from Palazzo et al. (2018; 2021) using the same hyperparameters as Li et al. (2021), Ahmed et al. (2021; 2022), and Bharadwaj et al. (2023).

# 5 RESULTS

Subject attention was confirmed through analysis of eye tracking data and experimenter observation. The results of the EEG analysis are shown in Table. 1.

## 6    DISCUSSION

Condition I replicates Spampinato et al. (2017), Kavasidis et al. (2017), and Palazzo et al. (2017; 2018; 2020a;b; 2021; 2024). As expected, it exhibits near perfect classification accuracy. Condition II differs from Condition I only in that the order of the blocks for the classes and the stimuli within the blocks differ across subjects. Nonetheless, the training and test samples for each class come from the same block, and therefore as expected, it also exhibits near perfect classification accuracy. Conditions III and IV differ from Conditions I and II in that the training and test samples for each class come from different blocks presented in different orders. Conditions III and IV differ from each other in that in Condition III the training samples all have class correlated with time across subject but the test samples do not, while in Condition IV the test samples all have class correlated with time across subject, but the training samples do not. Conditions III and IV exhibit essentially chance accuracy, in essence replicating the disputed results from Li et al. (2021) and Ahmed et al. (2022). Since Conditions III and IV differ from Conditions I and II only by assuring that training and test samples come from different blocks, which breaks the correlation between class order and time in the training set or test set, this demonstrates that the results of Spampinato et al. (2017), Kavasidis et al. (2017), and Palazzo et al. (2017; 2018; 2020a;b; 2021; 2024) crucially depend on this temporal confound.

It should be noted that while we release our dataset to the public, we emphasize that we do *not* suggest that it is appropriate to use as a benchmark for future work to measure ability to read out concepts from brain activity evoked by visual stimuli. Our dataset was explicitly constructed as a controlled experiment specifically to test the hypothesis that the protocol in Spampinato et al. (2017) involves a confound. As such, it repeated that experiment exactly, carefully varying only one controlled variable, namely stimulus presentation order. In contrast to the datasets used in Li et al. (2021), Ahmed et al. (2021; 2022), and Bharadwaj et al. (2023), the dataset used here is not truly randomized; other confounds may lurk in the current dataset. Indeed, it might measure only the ability to read out the general mind state of a subject flooded with images of the same class, not the ability to read out general naturally occurring perception.

## 7    CONCLUSION

We advise the community to stop using these confounded datasets and stop collecting new datasets with this flawed protocol. As is widely known in other experimental fields, only proper randomization of stimulus presentation order can avoid such temporal confounds (Ihrke & Behrendt, 2011; Abid et al., 2021; Mc Govern et al., 2017; Pollatsek & Well, 1995). There should be wide recognition within the community that all results based on these confounded datasets, or any other datasets collected with this non-randomized protocol, cannot provide accurate data and therefore cannot be trusted.

AUTHOR CONTRIBUTIONS

Removed for blind review.

ACKNOWLEDGMENTS

Removed for blind review.

ETHICS STATEMENT

Removed for blind review.

This work debunks nearly one hundred published papers whose results are based on the same confound: a correlation between stimulus class and temporal drift. This confound has been found in eighteen available EEG datasets. Just as with an inconsistent set of axioms one can prove anything, a confounded dataset can be used to support any claim, even ones that are false or absurd. That is what many recent publications based on this confound do: things like generating high fidelity renderings of images, or even 3D CAD models of objects, from EEG recordings.

A research community, knowingly or unknowingly, has discovered that one can use confounded datasets to churn out a plethora of flawed results without reviewers noticing. They have also discovered that one can collect new confounded datasets to churn out even more flawed results without reviewers noticing. The temptation to do this is so strong that the community continues to do so four years after details of the confound were published.

It is conceivable that the flaws in these datasets may be a driving factor behind their frequent reuse. When a dataset is severely confounded, it becomes relatively easy to achieve an extremely high accuracy, which can in turn be used to support sensational claims, and ultimately directs further attention to the dataset. In business, this phenomenon is referred to as "the bad money drives out the good money."

More prominent exposure of these flawed methods and consequent false results will allow resources wasted on continued use of these confounded datasets and flawed methods to be reallocated. The debunked work also causes direct ongoing harm:

- grant proposals can be rejected due to preliminary results not being competitive with results demonstrating falsely-inflated performance based on confounded data or faulty methods;
- manuscripts can be rejected for the same reason;
- grants can be awarded based on false pretenses
- manuscripts can be accepted for the same reason;
- degrees can be awarded for the same reason;
- resources can be wasted attempting to replicate the debunked results;
- resources can be wasted having people read and review flawed papers, and learn flawed methods; and
- because the debunked work relates to brain-computer interfaces—whose primary application is helping people with disabilities (*e.g.*, paralysis) interact with the world—the harm caused is not merely scientific but also medical, with disproportionate impact on people with disabilities.

## REPRODUCIBILITY STATEMENT

All code and raw data needed to replicate the results presented here will be released upon publication.

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
