# OpenReview forum: "A widely used protocol for EEG classification experiments leads to a confound"
_ICLR.cc/2026/Conference — Submitted to ICLR 2026_

### Official Review · Reviewer_TyXs · 2025-10-14

**Soundness:** 3
**Presentation:** 1
**Contribution:** 2
**Rating:** 2
**Confidence:** 4

**Summary:**

The paper focuses on temporal drift as a confound or 'embedded clock' in prominent EEG experimental designs. In particular, correlation of stimulus class with experiment timing. Although corrected experimental designs have been suggested, Palazzo et al. 2020 and 2024 claim the original confound is acceptable on several bases. The current paper rebuts this claim and highlights that the confounder remains.

**Strengths:**

The paper is well motivated and highlights an important problem in the literature (as well as problem in the peer-review system itself). The misapplication and misinterpretation of EEG data may also carry serious ethical concerns, e.g., if deployed in critical healthcare settings. For these reasons, it is very important that all such experimental designs are well-reasoned and understood.

**Weaknesses:**

I found the presentation and writing to be quite poor. On pages 2 and 3, there are huge numbers of in-text citations. It would be better to put this list in the appendix, for example, and save the limited real estate of the conference paper for more important points. On this point, the paper itself comes in at under 5 pages, with no appendices. This itself is not a showstopper for me: some of the best papers are short. However, if the quantity of content is limited, I expect the content in the main text to be information-dense and of very high quality. I do not find that it is, both in presentation and content.

In particular:
1. The paper primarily replicates and extends prior refutation studies with minor methodological variations, offering limited new theoretical or analytical insight.
2. Only six subjects were tested, which constrains generalisability and statistical power, especially given EEG’s high inter-subject variability.
3. While the study controls the temporal confound, other possible sources of bias (e.g., fatigue, attention drift, order effects beyond class blocks) are acknowledged but not experimentally tested.

**Questions:**

1. What anonymised information about the human participants did you record?
2. What informed consent and data protection procedures did you put in place?

**Details Of Ethics Concerns:**

There is no mention of the study's ethics board approval or an informed consent procedure for the human participants, nor of data protection procedures. This information may be in part of the content removed for anonymity -- but, if not, this presents an ethical concern, in my view.

---

> ### Author Response · Authors · 2025-11-21
>
> This manuscript is a direct response to Palazzo et al. (2024) claiming
> specific flaws in Li et al. (2021), Ahmed et al. (2021, 2022), and
> Bharadwaj et al. (2023). Its sole purpose is to show that the claims
> of Li et al. (2021), Ahmed et al. (2021, 2022), and Bharadwaj et
> al. (2023) are still valid and the work of Spampinato et al. (2017),
> Kavasidis et al. (2017), Palazzo et al. (2018, 2020a, 2020b, 2021,
> 2024), and the one hundred other papers are still flawed. Scientific
> debates are best won with simple arguments, not complex ones. The fact
> that it has "minor methodological variations, offering limited new
> theoretical or analytical insight" is a feature, not a bug, when it
> comes to correcting the scientific record.
>
> We chose six subjects to replicate the data collection of Spampinato
> et al. (2017) exactly. The differences between columns I/II and III/IV
> in Table 1 are so stark that they cannot be explained away by
> "generalisability and statistical power, especially given EEG’s high
> inter-subject variability".
>
> Re "other possible sources of bias (e.g., fatigue, attention drift,
> order effects beyond class blocks) are acknowledged but not
> experimentally tested": while these are indeed interesting issues, as
> stated above they are not necessary to address the sole issue of this
> paper, namely responding to Palazzo et al. (2024) and correcting the
> scientific record.
>
> Q1: none. Except that as described in the submission, we have eye
> tracking data for all subjects recorded during EEG recording to verify
> attentiveness. No additional data seemed relevant to our refutation
> purpose.
> Q2: yes this was approved by the IRB, removed for double-blind review.

---

### Official Review · Reviewer_5HxH · 2025-10-28

**Soundness:** 1
**Presentation:** 1
**Contribution:** 1
**Rating:** 0
**Confidence:** 4

**Summary:**

This paper contains similar elements as another paper submitted to this conference.

**Strengths:**

None

**Weaknesses:**

A similar paper has been submitted to this conference.

**Questions:**

none

---

> ### Comment · Reviewer_5HxH · 2025-11-27
>
> I would like to thank the authors for putting everything in a broader context.
>
> - If Li et al. (2021), Ahmed et al. (2021) and Bharadwaj et al. (2023) have shown that the dataset has been confounded. What is the added value of this (and other) papers?
>
> - Is it necessary to publish a high quantity amount (5) of papers in one conference venue to refute previous work? Why did you not target a very high impact journal or consolidate all findings in one high quality paper?
>
> with respect to this paper:
> - I understand that training on group A would have bad results on testing in on group B due to the confound (condition III), but would condition IV not result in larger accuracies?  Does the confound also has its impact during testing phase, even though training has been done on randomized blocks as the authors suggest.

---

### Official Review · Reviewer_C5VV · 2025-11-01

**Soundness:** 3
**Presentation:** 2
**Contribution:** 2
**Rating:** 4
**Confidence:** 3

**Summary:**

This paper presents a replication experiment demonstrating that a widely used block-design EEG classification protocol induces a temporal confound, linking the downstream class with time. It shows that classification accuracy depends heavily on this confound rather than true neural decoding. The paper critiques the continued use of flawed datasets and protocols in the field.

**Strengths:**

The concern identified in about temporal confounds in EEG classification experiments is apt and essential to communicate to the scientific community, because such confounds fundamentally undermine the validity of a large body of published EEG decoding research, potentially misguiding future methods development and applications.

Addressing this issue is critical for ensuring methodological rigor, truthful interpretation of results, and the ethical advancement of EEG-based neuroscience and brain-computer interface research.

The work is a replication of prior protocol with controlled manipulation of stimulus block order across multiple subjects, providing a thorough critique of defending arguments for block design, short sessions, and pooling subjects.

**Weaknesses:**

The language is generally direct but sometimes overly assertive in tone, bordering on confrontational regarding the ongoing use of flawed datasets in the community.

**Questions:**

Why have the authors not included similar analyses directly on the original datasets from Spampinato et al. or other widely used confounded datasets, to further validate the generalizability of their findings?

**Details Of Ethics Concerns:**

Agreeing with the concern raised earlier by reviewers, the articles 4793 and 4802 are very similar. The two papers strongly overlap in thematic concerns and call for randomized designs but differ in specific emphases: one paper is a detailed experimental replication and refutation targeting a widely used protocol and its dataset, while the second is a broader scope showing pervasiveness of temporal correlation across datasets and disproving filtering as a solution. Both could have been combined given the context and text including the citations are highly similar.

---

> ### Author Response · Authors · 2025-11-21
>
> W: We contend that strong language is necessary due to the egregious
> nature of the issue. The community continues to submit and publish
> over fifty new papers (including one submitted to ICLR 2025 and one
> submitted to ICLR 2026) despite publication of notice of the flaw once
> in CVPR and three times in TPAMI. Many authors (including the authors
> of the aforementioned submissions to ICLR 2025 and ICLR 2026) knowingly
> continue to submit. 4802 is a direct response to Palazzo et al. (2024)
> which denies the claims of Li et al. (2021), Ahmed et al. (2021,
> 2022), and Bharadwaj et al. (2023).
>
> Q: Because Spampinato et al. refuse to release their raw data to allow
> such analyses. (We have asked.) Authors of the other confounded
> datasets similarly do not release their data, despite promises in the
> paper to do so. (We have asked.)
>
> See above regarding plagiarism and dual submission.
>
> Indeed "differ in specific emphases: one paper is a detailed
> experimental replication and refutation targeting a widely used
> protocol and its dataset, while the second is a broader scope showing
> pervasiveness of temporal correlation across datasets and disproving
> filtering as a solution" is an argument that these merit being
> distinct manuscripts.
>
> "overlap in thematic concerns" is not reason to conflate the technical
> issued by merging into a single paper, especially given page limits.
>
> Re: "could have been combined": the strongest papers are ones that
> focus on making a single point irrefutably. The more one crams into a
> single paper, the more it weakens the paper by diffusing the
> focus. Since the community continues to publish and submit a large
> quantity of flawed work (even to ICLR 2025 and 2026), despite Li et
> al. (2021), Ahmed et al. (2021, 2022), and Bharadwaj et al. (2023),
> suggests that laser focus is prudent to direct the field towards a
> scientifically sound course.

---

### Official Review · Reviewer_pPTE · 2025-11-06

**Soundness:** 2
**Presentation:** 1
**Contribution:** 1
**Rating:** 2
**Confidence:** 2

**Summary:**

There seems to be a strong similarity between submissions 4793 and 4802. Both submissions point out a weakness of the experimental design of prior work studying visual stimuli decoding from EEG. While this reviewer agrees that randomized stimuli designs are critical in data collection to reduce the impact of confounding factors like temporal drifts often encountered in biosignals, it seems very likely that both articles originate from the same authors.

**Strengths:**

N/A

**Weaknesses:**

N/A

**Questions:**

N/A

**Details Of Ethics Concerns:**

To this reviewer, it seems likely that the authors submitted essentially the same work (submission 4793 and 4802) through different channels in hopes of achieving acceptance. While the authors might have a valid scientific critique, this method of presentation is not acceptable.

---

> ### Author Response · Authors · 2025-11-21
>
> There is no overlap between 4793 and 4802 except for §2, which is just
> a significance statement. Both 4793 and 4802 offer independent
> demonstration of the confound in the Perceive dataset (Spampinato et
> al. 2017). The technical details of the methods are completely
> different. Both are novel paradigms that have never been previously
> published. But they are both significant for the same reason: they
> refute the same body of over one hundred papers that use the Perceive
> dataset.
>
> 4793 provides evidence that all EEG data exhibits drift. It does this
> by demonstrating how that drift manifests itself as the ability to
> classify "classes" that aren't there, i.e., EEG recorded in the
> absence of stimuli. This was done on data collected by other people
> for other purposes, in three different labs.
>
> 4802 addresses a specific claim raised in Palazzo et al. (2024): that
> the work in Li et al. (2021), Ahmed et al. (2021, 2022), and Bharadwaj
> et al. (2023) is flawed due to long sessions. Submission 4802 repeats
> the data collection of Spampinato (2017) exactly in a controlled
> setting, varying a single parameter, namely block order. The design
> allowed training on one block order and then testing on either the
> same block order or a different block order. In the former, accuracy
> was nearly perfect. In the latter, accuracy dropped to chance. This
> conclusively demonstrates that the high accuracy of the former is due
> to a temporal correlation. Palazzo et al. (2024) specifically denied
> that high accuracy results from a temporal correlation and claimed
> that the chance accuracy reported by Li et al. (2021), Ahmed et al.
> (2021, 2022), and Bharadwaj et al. (2023) is due to long sessions and
> the randomized stimuli. This experiment shows this is not the case.
> Chance accuracy even results from short sessions with a block design
> when the temporal correlation is broken. This conclusively
> demonstrates that one hundred papers suffer from a confound: they work
> because of temporal correlation in the data and cease to work when
> that temporal correlation is broken.
>
> Both of these experiments are novel. This is an important contribution
> not only because it points out flaws in numerous prior papers, but
> also because it provide novel concrete methods for assessing whether
> other (potentially future) work is similarly flawed.
>
> Our five submissions are neither plagiarism nor dual submission. We
> cite all relevant prior publications and place all quotations in
> italics with citations. Each submission is completely disjoint in
> substantive content sharing only the significance statement because
> they refute that same large body of work. Upon acceptance, we would be
> happy for our submission to cross cite each other, and include only a
> single significance statement. But that is not possible for
> double-blind review when the same reviewers cannot see all five.
>
> We did not "submit[ted] essentially the same work (submission 4793 and
> 4802) through different channels in hopes of achieving acceptance".
> These are distinct papers making distinct claims through distinct
> experiments offering distinct refutations of the same large body of
> flawed work.

---

### Author Response · Authors · 2025-11-21
**Historical Background and Significance**

To understand this work's significance, consider this brief historical
overview.

Spampinato et al. (2017) introduced a block-designed dataset
("Perceive") and methods that claim to achieve extremely high accuracy
decoding stimulus image class from EEG recordings. This was amplified
by follow on papers (Kavasidis et al. 2017, Palazzo et al. 2018,
2020a, 2020b, 2021), many of which claim to do things like reconstruct
stimulus images from EEG recordings. Further, Tirupattur (2018) does
this with a fresh dataset (Kumar 2018) that has the same block design.

Li et al. (2021) debunked all of the above, demonstrating that the
Perceive dataset suffers from a block confound. EEG exhibits drift,
encoding a clock in the signal. Since Perceive was collected with all
and only stimuli of the same class being temporally adjacent, the
classifier can mistakenly classify the clock/drift instead of the
stimulus-related EEG response. Follow on papers (Ahmed et al. 2021,
2022, Bharadwaj et al. 2023) added novel independent confirmation of
the results of Li et al. (2021).

Despite this, Palazzo et al. (2020b, 2021, 2024) continue to argue
that their dataset is valid. At this point, there are over one hundred
papers that use the Perceive dataset, the Kumar (2018) dataset, or
other datasets that suffer from the same block confound. Many new
datasets have been collected with this same block confound, some of
which are becoming widely used. The vast majority of these were
published after the confound became known (Li et al. 2021). Some of
these are unaware of the confound. Others are aware, but dismiss it,
often based on the argument of Palazzo et al. (2020b, 2021, 2024).

That argument is what this manuscript refutes.

This confound has been extensively debated on blogs like reddit, yet
that too has not stopped the extensive publication of flawed work.

There are three distinct levels of severity of this issue, which
progressively support greater need for continued publication:

 1. Many authors are unaware of the confound, despite the fact that it
    was published in prominent venues (e.g., TPAMI, CVPR, NeurIPS) and
    continue to publish flawed work

 2. While many authors are aware of the confound, they nonetheless
    ignore the warning and continue to publish flawed work.

 3. Some authors dismiss the confound and actively argue for the
    community to continue to employ flawed methods.

---

### Author Response · Authors · 2025-11-21
**Re: Concerns about duplicate submissions**

We submitted five manuscripts to ICLR 2026. To summarize:

  4790: Palazzo (2020b, 2021) introduces a method that claims to
        jointly train two mappings, from EEG and images, to a common
        embedding space. We debunk central claims about this
        embedding. We do this both for the confounded dataset and
        nonconfounded datasets.

  4793: Palazzo et al. (2020b, 2021, 2024) claim that their dataset
        does not suffer from drift. We show that three other datasets,
        all collected by different people in different labs, suffer
        from drift, demonstrating that drift is unavoidable with EEG.

  4796: Palazzo et al. (2020b, 2021) produce activation maps and claim
        that these are consistent with neuroscience knowledge. We
        debunk this claim.

  4802: Palazzo et al. (2024) further claim that their dataset is
        valid by arguing that the experiments in Li et al. (2021),
        Ahmed et al. (2021, 2022), and Bharadwaj (2023) were
        improperly conducted. We repeat the experiment in Spampinato
        et al. (2017) exactly, in a controlled fashion, where the only
        thing varied is block order. this conclusively demonstrates
        that Spampinato et al. (2017), Kavasidis et al. (2017),
        Palazzo et al. (2018, 2020a, 2020b, 2021, 2024), and the one
        hundred other papers are flawed.

  4804: Palazzo et al. (2024) makes numerous false statements about
        Li et al. (2021), Ahmed et al. (2021, 2022), and Bharadwaj (2023).
        We correct those statements.

These are all independent. There is no duplicate substantive material
between these five submissions and Li et al. (2021), Ahmed et al.
(2021, 2022), and Bharadwaj (2023). While they all comment on the same
body of flawed work, they each introduce and discuss distinct
technical issues and make distinct contributions.

We included §2 Significance in all five manuscript. While largely the
same text, this is not the technical contribution of each respective
manuscript. It solely serves to highlight the significance of the
specific technical contribution in each individual manuscript, namely
that each offers independent refutation of one hundred papers. This is
important, because even if one were to remedy one of the flaws, many
others remain, and a large and growing corpus of work remains flawed.
Further, it is conceivable that in the future, a paper might suffer
from one flaw but not the other, yet it would still be invalid.

---

### Author Response · Authors · 2025-11-21
**Re: Public debate**

Several reviewers commented that public debate of this issue is
inappropriate. We realize that this may be unconventional and uncommon
in the ML community. But it is common in most other scientific fields
(e.g., Brain and Behavioral Science, Psycoloquy, ...). Public debate
through publication is the well-established method for arriving at
scientific truth. Schaeffer (2025) have argued that a mechanism for
publishing critiques and refutations is sorely lacking in ML.

The vast majority of the reviews of all five of these submissions
focus on the fact that they are unconventional. Essentially none of
the reviews discuss any technical flaws in these submissions. We
would be happy to discuss and address any technical flaws.

---

### Author Response · Authors · 2025-11-21
**Specific relevance and significance to ICLR and the ML community**

It is important, if not imperative, for the community to publish this
work. Without it, the community continues to submit and publish more
flawed work at a growing rate. Fifty new papers papers have been
published since the flaw was first reported in prominent venues: once
in CVPR (Ahmed et al. 2021) and three times in TPAMI (Li at al. 2021,
Ahmed et al. 2022, Bharadwaj et al 2023).

Some recent flawed work has been published even by the ML community in
top ML venues, despite awareness of the issue: Liu et al. (2024) in
NeurIPS collects a new dataset that suffers from the block
confound. While the authors cite Li et al. (2021) and Ahmed et al
(2021), they fail to appreciate (or maybe hide the fact) that their
work is confounded.

Some recent flawed work has even been submitted to ICLR 2025 (and
apparently resubmitted to ICLR 2026 despite reviewer warnings). It
appears that even the reviewer pool of ICLR is unaware of the severity
of the confound.

https://openreview.net/forum?id=ejVuTFFkl6&noteId=zafmRtlFw1

collects a new dataset that suffers from the block confound. While the
authors again cite Li et al. (2021), they incorrectly claim that their
dataset does not suffer from the confound. All four of the reviewers
point this out. While this submission was rejected, three of the
reviewers rated it as "Soundness: 3: good" and two of the reviewers
rated it as "Contribution: 3: good".

The apparent resubmission (18265) to ICLR 2026 again cites Li et
al. (2021) and again incorrectly claims that their dataset does not
suffer from the confound. Again, three of the four reviewers point out
that this work suffers from the block confound. And again, two of the
reviewers rate this as "Soundness: 3: good", one of the reviewers
rates this as "Contribution: 3: good", and one even rates this as
"Contribution: 4: excellent" and recommends acceptance.

We have a simple question for the reviewers, area chairs, and program
chairs: If one cannot publish refutations like this in ICLR, how else
do you propose we address the fact that there is a large and growing
body of flawed work being published?

---

### Meta-Review · Area_Chair_DDZq · 2026-01-01

**Summary:**

Reviewers’ concerns centred on research-integrity issues (substantial overlap with closely related submissions and possible salami-slicing), poor presentation and clarity, and limited incremental contribution, with questions about generalisability and validation despite addressing a known confound.

**Reviewer Concerns:**

The rebuttal clarified the authors’ intent and positioning of this paper relative to their other submissions and provided justifications regarding tone and experimental design. However, the core reviewer concerns remain outstanding, including the research-integrity/overlap issue, poor clarity and presentation, and limited incremental contribution and generalisability. As a result, the rebuttal did not materially change the substance of the reviewers’ negative assessments.

**Reviewer Scores:**

Given that all reviewers initially gave negative scores and only one participant in the discussion (and further confirmed the negative score), and that the rebuttal did not resolve the fundamental concerns, I do not believe any reviewer would have materially increased their score.

---

### Decision · Program_Chairs · 2026-01-26

Reject